# Locally Activated Gated Neural Network for Automatic Music Genre Classification

**Zhiwei Liu** [1,*], **Ting Bian** [2] **and Minglai Yang** [2,*]

1 School of Electronic and Electrical Engineering, Shanghai University of Engineering Science, Shanghai 201620, China
2 School of Railway Transportation, Shanghai Institute of Technology, Shanghai 201418, China
* Correspondence: M020119111@sues.edu.cn (Z.L.); yangminglai@sit.edu.cn (M.Y.)

**Abstract:** Automatic music genre classification is a prevailing pattern recognition task, and many algorithms have been proposed for accurate classification. Considering that the genre of music is a very broad concept, even music within the same genre can have significant differences. The current methods have not paid attention to the characteristics of large intra-class differences. This paper presents a novel approach to address this issue, using a locally activated gated neural network (LGNet). By incorporating multiple locally activated multi-layer perceptrons and a gated routing network, LGNet adaptively employs different network layers as multi-learners to learn from music signals with diverse characteristics. Our experimental results demonstrate that LGNet significantly outperforms the existing methods for music genre classification, achieving a superior performance on the filtered GTZAN dataset.

**Keywords:** deep learning; music genre classification; gated network; convolutional neural network

## 1. Introduction

Automatic music genre classification is a well-established task in the field of machine learning [1]. Its objective is to accurately classify given music tracks into a specific genre or set of genres [2,3]. The potential applications of music genre classification in music recommendation systems [4] and music streaming services [5] have led to extensive research in this area.

Music genre classification typically involves two steps: acoustic feature extraction and classification. As a critical component of music genre classification, acoustic feature extraction extracts meaningful characteristics from music tracks. Traditional music features include loudness, rhythm [1], beat [1,6], a zero-crossing rate [6], and Mel-frequency cepstral coefficients (MFCCs) [7–9]. In addition, researchers have also explored spectrograms based on Fourier transform [10,11], wavelet transform [12], or constant-Q [13] transform, which contain rich time-frequency information (e.g., temporal information, periodic beat, rhythm, etc.) and can achieve a more satisfactory performance.

The other critical component of music genre classification is designing classification algorithms to handle acoustic features. Classic machine learning algorithms include statistical methods, such as naive Bayes classifiers [14], random forests [15], and support vector machines (SVMs) [7,8]. Meanwhile, some studies have revealed that classic machine learning models may not be suitable for large-scale data with diverse data distribution [4,5]. With the advancement of deep learning [16] and computing resources, classification algorithms based on deep neural networks continue to grow in popularity. Most researchers choose recurrent neural networks (RNNs) [17], convolutional neural networks (CNNs) [18,19], or Transformers [9] as the classification backbone for music genre classification. Such deep learning-based methods could capture the latent information in acoustic features (e.g., timbre information and semantic information ), thus achieving a better performance in real-world applications.

However, there are still some limitations to the existing methods. Based on the analysis of the music signals, we observe that the genre of music is a very broad concept [20]. Music tracks belonging to the same genre may have diverse acoustic characteristics, such as the rhythm and beats. Figure 1 illustrates the significant variation in spectrograms of music tracks belonging to the "blues" genre.

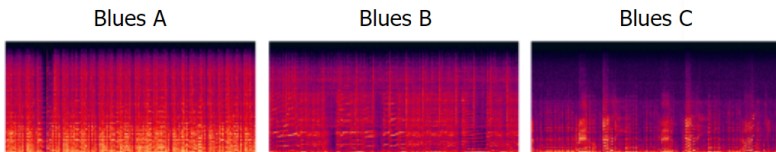

**Figure 1.** The illustration of the spectrogram differences in blues music. For all three sub-figures, the *x*-axis ranges from 0 to 30 s, while the *y*-axis ranges from 0 to 11,025 Hz.

The drawback of the existing methods is that they are not good at dealing with such diverse data distributions with large intra-class differences. To correctly classify Blues A and Blues C into the same genre, the model needs to further capture deep latent information. However, when the amount of data is insufficient for "deeper digging," it can lead to false inductive bias and adversely affect the classification accuracy.

In this work, we specifically address the difficult problem of music genre classification with large differences in intra-class data distribution. We propose the **l**ocally activated **g**ated neural network (LGNet) for music genre classification. LGNet includes multiple multi-layer perceptrons (MLPs) [21] and a gated routing layer [22] on the backbone of the classification network. We design several MLPs and employ them as multi-learners, focusing on knowledge in different aspects. The gated routing layer is applied to determine the allocation of inputs by calculating the matching degree between the input representations and each MLP. When the model encounters a certain sample, only the MLP with a high matching degree is activated, while the remaining MLPs are deactivated. In this manner, the model can adaptively allocate the most appropriate network layer according to the input samples. Taking Figure 1 as an example, LGNet can dispatch the MLP layer specialized in processing periodic beats and rhythms to Blues A while dispatching the MLP layer specialized in processing low-frequency line spectra to Blues C. Such a locally activated structure introduces more parameters for complex modeling, and due to its locally activated property, additional parameters do not reduce the training and inference speed. Thereby, LGNet can alleviate the problem of large intra-class differences in an efficient manner.

According to our experimental results and analysis, we demonstrate that LGNet is highly effective for music genre classification with large intra-class differences. The locally activated gated network can achieve a satisfactory performance on the GTZAN dataset [1]. The contributions of our work are summarized as follows:

- We reveal the intra-class differences problem in music genre classification, which impedes the progress of recognition performance.
- We propose the locally activated gated network, which can adaptively dispatch the most appropriate network layer based on inputs.
- Our experimental results demonstrate that LGNet outperforms the existing methods on the filtered GTZAN dataset.

## 2. Related Works

### 2.1. Classic Machine Learning

The proposal for a fully automatic music genre classification system was first put forth by Tzanetakis and Cook [1]. They presented three feature sets to represent the timbral texture, rhythmic content, and pitch content and trained statistical pattern recognition classifiers using real-world audio collections. The GTZAN dataset they released has become the benchmark dataset for most subsequent work in this area. Since then, various algorithms

based on traditional machine learning have been proposed. Xu et al. [6] employed the multi-layer classifier based on support vector machines to replace traditional Euclidean distance-based methods and other statistic learning methods. Patil and Nemade [8] used acoustic features such as the MFCC vector, chroma frequencies, spectral roll-off, spectral centroid, and zero-crossing rate and combined SVM and K-NN (K Nearest Neighbor) algorithms to accomplish the classification task. Chaudhury et al. [3] investigated naive Bayes, decision trees, logistic regression, and random forest for the classification of music genres.

### 2.2. Deep Learning

With the advent of deep learning algorithms and computing resources, recognition algorithms based on deep neural networks have become increasingly prevalent in this field. Liu et al. [2] utilized a middle-level learning feature interaction method based on the convolution neural network; Abeßer and Müller [23] captured the music theory of Jazz by a recently proposed U-net deep neural network architecture; Zhuang et al. [24] designed a Transformer classifier to analyze the relationship between different audio frames; and Khasgiwala and Tailor [9] evaluated the performance of classification systems based on the novel Vision Transformer, RNN-LSTM (Long Short-Term Memory), and CNN-based architecture using MFCCs as the acoustic feature. As reported in the literature, deep learning-based methods can generally achieve a more satisfactory performance than those based on classic machine learning.

In addition, as a data-driven technique, the success of deep learning methods cannot be separated from the construction of large-scale datasets. In addition, the performance of a classification system depends on the quality and scale of the dataset. There are many free and open source datasets available for music genre classification, including the GTZAN Genre Collection (GTZAN), Free Music Archive (FMA), and Million Song Dataset (MSD). The GTZAN Genre Collection is a widely used dataset for music genre classification and includes pre-extracted features, such as Mel-frequency cepstral coefficients (MFCCs) and chroma features. The Free Music Archive is a large collection of free, legal music that has the associated metadata, such as the artist, album, and genre. The Million Song Dataset is a collection of audio features and metadata for a million contemporary popular music tracks. The MSD includes audio features, such as the pitch, timbre, and rhythm, as well as the metadata, such as the artist, year of release, and popularity. These datasets contribute to an open community and provide sufficient training data for data-driven deep learning models.

## 3. Materials and Methods

In this section, we give a detailed introduction to our LGNet, including our used acoustic features, the neural network backbone, and the complete training flow.

### 3.1. Acoustic Feature Extraction

According to the literature, the acoustic features based on the spectrogram contain rich time-frequency information. In recent years, applying neural networks to learn from the time-frequency spectrograms has become the most popular paradigm in automatic music genre classification. In this work, we perform three spectrogram-based acoustic features to verify the generalizability of our proposed strategy. We describe the feature extraction process in detail as follows:

STFT spectrogram: First, the input signal is windowed (e.g., using the Hanning window function) to reduce the effect of spectral leakage, which can cause the spectral components of the signal to spread beyond their actual frequency range. Next, the windowed signal is partitioned into short segments with overlapping (this operation is referred to as "framing"). For each segment, the Fourier transform is computed, resulting in a frequency-domain representation of the segment. Finally, the modulo frames are assembled into the STFT spectrogram, with time on the x-axis and frequency on the y-axis. The length of the Fourier transform determines the frequency resolution of the spectrogram, while the length of the segment determines its temporal resolution.

Mel spectrogram: The Mel spectrogram is a variation of the traditional spectrogram that emphasizes frequencies according to the perceptual characteristics of the human ear. As mentioned above, we can obtain the FFT spectrum by windowing, framing, and Fourier transform. The next step involves applying Mel filter banks to the spectrum. The Mel filter bank is a set of triangular bandpass filters that are spaced according to the Mel scale, which is a non-linear perceptual scale of frequency (as depicted in Equation (1)). Each filter in the bank is centered at a particular Mel frequency and has a bandwidth that varies according to the Mel scale. Finally, the filtered spectrum is transformed using a logarithmic scale to obtain the Mel spectrogram.

$$Mel(f) = 2595 \times log_{10}(1 + \frac{f}{700}) \qquad (1)$$

CQT spectrogram: Constant-Q transform (CQT) is a popular tool for analyzing non-stationary signals with varying frequency content over time, such as musical signals. After obtaining the FFT spectrum, the CQT is computed by convolving the FFT spectrum of each windowed frame with a bank of bandpass filters that are logarithmically spaced in frequency (CQT kernel). Denote the maximum or minimum frequency to be processed as $f_{max}, f_{min}$. $f_k$ represents the frequency of the *k-th* component, and $b$ is the number of spectral lines contained in an octave (e.g., $b = 36$). $f_k$ can be formalized as

$$f_k = 2^{\frac{k}{b}} f_{min} \qquad k = 0, 1, \ldots \lceil b \cdot log_2(\frac{f_{max}}{f_{min}}) \rceil - 1 \qquad (2)$$

Then, the magnitude of the filtered spectrum is used to represent the CQT spectrogram. CQT obtains different frequency resolutions by using different window widths so that the frequency amplitude of each semitone can be obtained.

### 3.2. Neural Network Backbone

In this work, we adopt the deep residual network (ResNet-18) [25] as our network backbone. As shown in Figure 2, ResNet begins with convolving and pooling the input spectrogram to extract low-level features. This is followed by four basic blocks consisting of multiple convolutional layers, each with the skip connection that allows the input to bypass the block and flow directly to the output.

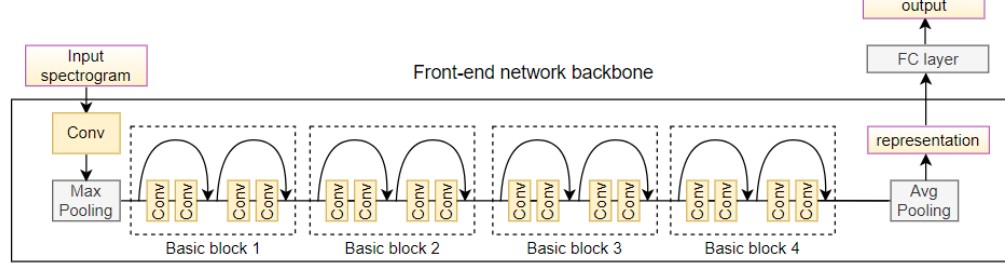

**Figure 2.** The framework of network backbone—ResNet18. "Conv" represents the convolutional layer (yellow box) and "FC" represents the fully connected layer. For brevity, we omit the ReLU and Batchnorm layers in the figure.

The basic block in ResNet-18 is called a "residual block", which is designed to solve the problem of vanishing gradients in deep neural networks. The residual block takes inputs and passes them through a series of convolutional layers, batch normalization, and activation functions—ReLU (we omit the ReLU and batch normalization in Figure 2 for brevity). The outputs from this series of layers is then added to the inputs, creating a "shortcut connection". The purpose of the shortcut connection is to preserve the gradient information from the inputs, which can be lost as it passes through the convolutional layers. By adding the inputs to the outputs, the residual block allows the network to learn residual functions, which are easier to optimize than the original functions. The convolutional

layers in the basic blocks have a filter size of $3 \times 3$ and a stride of 1, with padding used to keep the spatial dimensions of the inputs and outputs the same. The batch normalization layer normalizes the outputs from the convolutional layers, helping to reduce the effects of internal covariate shift and improve the stability and speed of the training process. The activation function used in the residual block is a rectified linear unit (ReLU), which helps to introduce non-linearity into the model and increase its capacity to learn complex representations.

ResNet also includes several pooling layers that downsample the feature maps to reduce the dimensionality of the input. The last adaptive pooling layer will pool the feature map into $512 \times 1 \times 1$, and the representation vector can be obtained after the flatten operation. The classic ResNet is followed by a fully connected layer to output the classification result, while our LGNet uses this representation for subsequent locally gated activation and assignment. We refer to this ResNet-based network architecture that transforms the input spectrograms into one-dimensional representations as "Front-end network backbone".

### 3.3. Framework of Locally Activated Gated Network

Figure 3 illustrates the framework of our proposed LGNet and its comparison with the classic classification network. Our proposed LGNet shares the same front-end network backbone as the classic classification network. Classic classification network directly feeds the representations into the linear fully connected layer and outputs the predictions. Such a method is not good at dealing with data distributions with large intra-class differences.

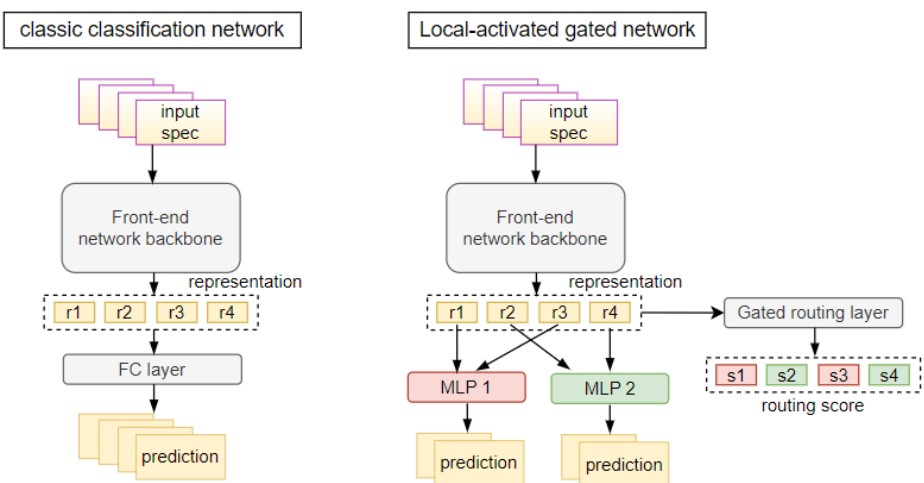

**Figure 3.** The framework of our proposed LGNet and its comparison with baseline classification network. "FC" represents fully connected layer, while "spec" represents spectrograms. This figure takes a batch with batch size = 4 as an example, the models are fed four independent samples, and output their corresponding predictions.

For our proposed LGNet, we feed the representations into the gated routing network to compute the routing score. Denote the batch size as n, input spectrograms as $x_i$, and the corresponding label as $y_i$ ($i$ = 1, 2…n). The front-end network takes $x_i$ as input and outputs the representations $r_i$. We use a linear gated routing layer $G(\cdot)$ to learn the latent characteristics contained in the representations, and then output the routing score $s_i = G(r_i)$. Next, the activation function transforms the routing scores into the routing probabilities $p_i$, whose value represents the probability of being assigned to the corresponding MLP layer. Take a structure composed of two MLPs as an example, when $p_i$ = (0.75, 0.25), the representation will be sent to the first MLP layer. The model computes routing probabilities for all representations and assigns them to the best-matched MLP layer based on $p_i$. MLP layers finally make predictions on the samples assigned to them.

Additionally, in order to pursue the stability of the model, we fix the parameters of the gated routing network after training for several epochs so that they will no longer be updated. This prevents changes in the parameters of the gated routing network from causing the allocation strategy to remain unfixed. The overall training flow for LGNet is detailed in Algorithm 1.

---

**Algorithm 1:** The overall training flow for LGNet.

---

**Data:** training samples $X$ and label $Y$.
**Input:** convolutional layers $f(\cdot)$; gated routing network $G(\cdot)$; MLP layers $L(\cdot)$;
          number of MLP layers $\mathcal{N}$; activation function $\sigma(\cdot)$; max epochs $E$; routing
          early stop epoch $\tau$.

1   **while** *epoch* <$E$ **do**
2      sample batches $(x_i, y_i) \sim (X, Y)$.
3      **for all** $(x_i, y_i)$ **do**
4          obtain music representations $r_i \leftarrow f(x_i)$.
5          compute routing scores $s_i \leftarrow G(r_i)$.
6          obtain routing probs $p_i \leftarrow \sigma(s_i)$.
7          obtain activated MLP id $Z_i \leftarrow argmax_i(p_i)$.
8          compute logits $logits_i \leftarrow L_{Z_i}(r_i)$.
9          compute loss $\mathcal{L}^i = Crossentropy(logits_i, y_i)$.
10      **if** *epoch* <$\tau$ **do**
11          update layers $f(\cdot), G(\cdot), L(\cdot)$ with loss $\sum_i \{\mathcal{L}^i\}$.
12      **else**
13          update layers $f(\cdot), L(\cdot)$ with loss $\sum_i \{\mathcal{L}^i\}$.

---

## 4. Experiment Setup

### 4.1. Dataset

We choose filtered GTZAN [1] as the dataset for this task, which contains 1000 tracks of 30-second length. There are 10 categories of music genres in GTZAN (blues, classical, country, disco, hip-hop, jazz, metal, pop, reggae, and rock), and each genre contains 100 tracks with a sampling rate of 22,050 Hz. In consideration of several annotation errors on the dataset, we adopt the "fault-filtered" split [26] to minimize the impact of error labels. For the fault-filtered dataset, the total number of audio clips is reduced to 930 as a result of filtering out mislabeled samples. The training, validation, test set, respectively, include 443, 197, 290 tracks. Given that it possesses more precise labels, we believe the fault-filtered split can better reflect the performance of the recognition model than the original split.

### 4.2. Baseline Methods

In this work, we perform several widely used pattern recognition techniques as baseline methods, including naive Bayes, Support Vector Machines (SVM), Long Short-Term Memory (LSTM), Bidirectional LSTM (Bi-LSTM), fully convolutional networks (FCN), FCN-LSTM, and residual networks (ResNet).

Naive Bayes is a probabilistic algorithm based on Bayes' theorem, which assumes that features are conditionally independent given the class label. SVM is a powerful algorithm that constructs a hyperplane to separate different classes in the feature space, often using a kernel trick to implicitly map the data to a high-dimensional space. LSTM is a type of recurrent neural network (RNN) that utilizes a memory cell to selectively forget or remember information over time, allowing it to handle long-term dependencies in sequential data. Bi-LSTM is an extension of LSTM that includes a backward pass through the sequence, enabling the network to capture information from both past and future inputs. FCN is a neural network architecture that consists entirely of convolutional layers. FCN-LSTM is a hybrid model that combines the strengths of FCN and LSTM and has been

shown to effectively model spatio-temporal data. Finally, ResNet is a deep neural network architecture that utilizes residual connections to enable the training of very deep networks and has achieved promising performance on a wide range of computer vision tasks and audio pattern recognition tasks.

### 4.3. Parameters Setup

During framing, this work sets the frame length as 30 ms and the frameshift as 15 ms. In addition, we set the number of Mel filter banks to 128 as default. During training, we use the AdamW optimizer with a weight decay of $1 \times 10^{-5}$. The initial learning rate is set to $5 \times 10^{-4}$. All models are trained for 300 epochs on a single V 100 GPU. The batch size is set to 64 as default.

## 5. Results and Discussion

### 5.1. Main Results

Table 1 presents the results for a music genre classification task, where different methods are evaluated with three different acoustic features. First, we compare the performance of many existing classification models, with ResNet performing the best. So, we chose it as the backbone of our network. Then, the experiments show that our proposed LGNet has overwhelming advantages over the other models. Our LGNet uses a similar network backbone as ResNet, on which only a few simple linear layers are added. Our strategies can bring a 6.66% (CQT) to 8.16% (Mel) performance improvement. In addition, we find that the number of MLP layers also has an impact on the results. The related research is detailed in Section 4.3.

**Table 1.** Main results on the filtered GTZAN dataset. All reported results are the accuracy on the test set.

| Methods | Accuracy (STFT) | Accuracy (Mel) | Accuracy (CQT) |
|---|---|---|---|
| Naive Bayes | 50.48 | 54.45 | 51.90 |
| SVM | 62.25 | 64.36 | 64.99 |
| LSTM | 60.14 | 61.44 | 60.79 |
| Bi-LSTM | 61.48 | 62.01 | 61.41 |
| FCN | 73.55 | 74.39 | 68.86 |
| FCN-LSTM | 72.59 | 73.70 | 71.28 |
| ResNet | 74.60 | 74.55 | 72.51 |
| LGNet-2MLP | 82.43 | 82.36 | 79.17 |
| LGNet-4MLP | 82.24 | 82.71 | 79.55 |

In addition, we plot a confusion matrix to further illustrate the detailed performance of our proposed LGNet on each genre. For the confusion matrix, darker colors indicate that the model has a greater probability of making corresponding predictions. In Figure 4, we observe that classic music and metal music are very well classified, while music with the genre of country and rock is harder to distinguish. Overall, the performance of the automatic classification system based on LGNet is satisfactory; it can make correct predictions for most of the music tracks.

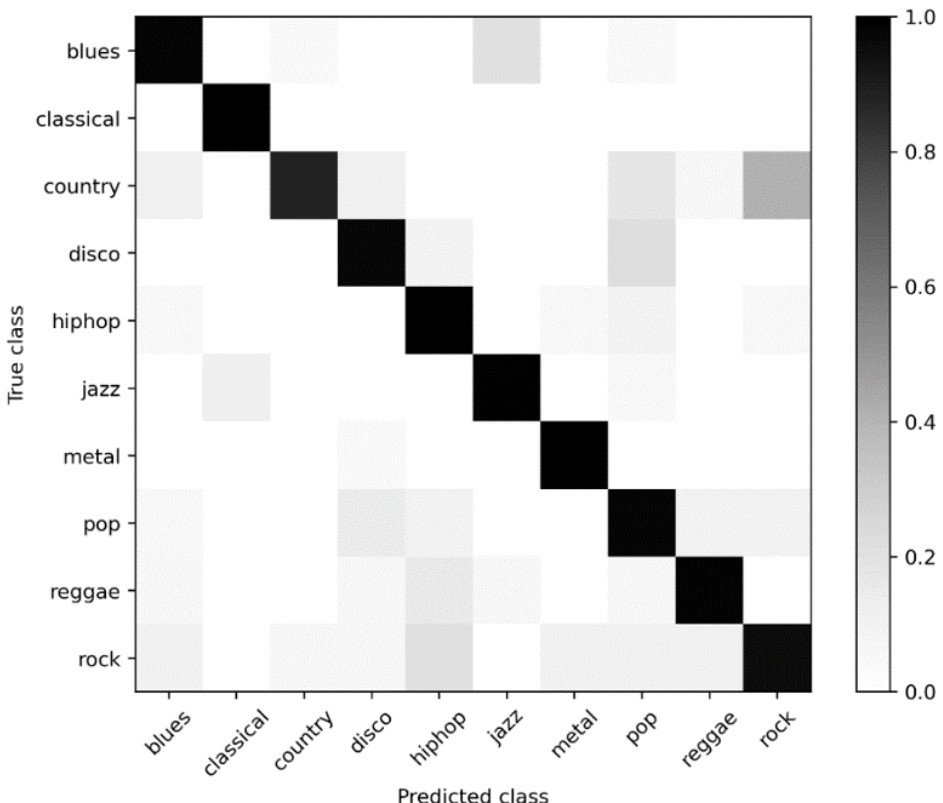

**Figure 4.** The confusion matrix for our best LGNet model (accuracy = 82.71%).

Now, let us turn our attention to acoustic features. We observe that models based on the Mel spectrogram outperform models based on other acoustic features in most cases. It shows that applying the filter bank with a high resolution at low frequencies is an effective method. In addition, the STFT spectrogram is also a well-behaved feature that sometimes could outperform the Mel spectrogram (74.60% vs. 74.55% and 82.43% vs. 82.36%). We also observe that models based on CQT spectrograms could not achieve a competitive performance. According to our inference, CQT spectrograms contain more frequency bins, including some less informative frequency bins that are not discriminative for classification.

*5.2. Selection of Activation Function*

Next, we investigate the selection of the activation function after the routing layer. To normalize the routing probability values, we convert the routing score to the routing probability by the activation function. There are two functions to choose from:

$$Softmax(x) = \frac{e^{x_i}}{\sum_{j=1}^{n} e^{x_j}} \tag{3}$$

$$Sigmoid(x) = \frac{1}{1 + e^{-x}} \tag{4}$$

As illustrated in Table 2, it is obvious that choosing the Softmax function is the best option. This is because Softmax combines the normalization of all the input values, and the output result is the correlation between the probabilities. For the sigmoid function, it cannot reflect the correlation between different probabilities, and because the gradient of the part far away from 0 is small, it is prone to gradient disappearance.

**Table 2.** Selection of activation function. We uniformly choose LGNet with 4 MLPs as our model.

| Methods | Accuracy (STFT) | Accuracy (Mel) | Accuracy (CQT) |
| --- | --- | --- | --- |
| Linear | 76.32 | 77.93 | 75.47 |
| Softmax | 82.24 | 82.71 | 79.55 |
| Sigmoid | 81.13 | 81.49 | 79.44 |

*5.3. Number of MLP Layers*

In this subsection, we explore the influence of choosing the appropriate number of MLP layers. The number of MLP layers has a strong correlation with the number of model parameters and the degree of model sparsity.

According to our experiments, in Table 3, choosing a moderate number of MLP layers (four layers) brings the best results. It is obvious that more MLP layers mean more parameters and a stronger ability to learn from complex data distributions. When the number of MLP layers is small, although LGNet can still bring obvious improvements compared to the original structure, its advantages have not been fully stimulated. When there are too many MLP layers (such as 16), the data allocated to each MLP layer is proportionally reduced, which may cause the MLP layers to not be fully trained. For example, if there are 1000 training samples in the dataset, each MLP layer of LGNet-4MLP can be allocated 250 training samples on average, while each MLP layer of LGNet-16MLP may only be allocated 62.5 training samples on average. It can cause models with too many MLP layers to suffer from overfitting due to the limited allocated data. The degradation of the model performance of LGNet-16MLP in Table 3 confirms our conclusion.

**Table 3.** LGNet with different MLP layers.

| Methods | Accuracy (STFT) | Accuracy (Mel) | Accuracy (CQT) |
| --- | --- | --- | --- |
| LGNet-2MLP | 82.43 | 82.36 | 79.17 |
| LGNet-4MLP | 82.24 | 82.71 | 79.55 |
| LGNet-8MLP | 82.20 | 82.59 | 79.67 |
| LGNet-16MLP | 81.71 | 81.46 | 78.90 |

## 6. Conclusions

This paper focuses on automatic music genre classification and reveals that the current methods are not good at dealing with diverse data distributions with large intra-class differences. Based on the issue, we propose an effective and parameter-efficient structure—the locally activated gated neural network. We employ multiple MLPs and employ them as multi-learners, focusing on knowledge in different aspects. In addition, our LGNet can adaptively allocate the most appropriate network layer according to the gated routing layer. According to our experimental results and analysis, we demonstrate that LGNet is very effective for music genre classification. It can achieve a superior performance on the filtered GTZAN dataset.

In the future, we plan to dig deeper into the potential of gating networks based on such local activations. We want to try a more complex routing layer and introduce an attention mechanism to control the weight of different MLP layers. Meanwhile, we are interested in exploring whether there is a more optimal alternative to the structure of the MLP layers.

**Author Contributions:** Conceptualization, Z.L.; methodology, Z.L.; validation, Z.L. and T.B.; investigation, Z.L. and T.B.; writing—original draft preparation, Z.L.; writing—review and editing, M.Y.; visualization, Z.L. and T.B.; supervision, M.Y. All authors have read and agreed to the published version of the manuscript.

**Funding:** This research received no external funding.

**Institutional Review Board Statement:** Not applicable.

**Data Availability Statement:** Our used GTZAN dataset is available at https://www.kaggle.com/datasets/andradaolteanu/gtzan-dataset-music-genre-classification. In addition, our train, validation, and test split is available at https://github.com/jongpillee/music_dataset_split/tree/master/GTZAN_split.

**Conflicts of Interest:** The authors declare no conflict of interest.

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
