# Peer review of "Locally Activated Gated Neural Network for Automatic Music Genre Classification"

_applsci, doi:10.3390/app13085010_

Round 1

Reviewer 1 Report

The paper presents a method to improve Genre Classification by tackling the problem of intra-class variability of data through a gating mechanism. The paper is very compact, clear, and the method seems to be efficient. There are not many remarks from my side, except the description of the data used can be improved. It is not clear how many songs are now actually used for training and for testing. First it seems there are 1000 songs used, but then the authors mention 443 songs and mention "fusion training set", which i don't know what that is. Please state clearly what the train/eval/test splits are. 

Author Response

We apologize for not explaining it clearly in the original paper. The original GTZAN dataset has 10 classes with 100 30-second audio clips in each class, and the whole dataset has 1000 audio clips. However, due to the rough construction of the dataset, there are some mislabels. Therefore, there are related works [1] that manually remove all mislabeled audios in the dataset and divide it into a fault-filtered split. For the fault-filtered split, the training set includes 443 samples, the validation set includes 197 samples, and the test set includes 290 samples. We modified the relevant parts of the original paper.

[1] Deep Learning and Music Adversaries, Kereliuk et al. 2015 Arxiv

Reviewer 2 Report

General remark: the article presents an interesting approach to classifying musical genres. The paper is easy to follow, the results are presented coherently, and the experiments conducted are discussed quite thoroughly. However, a few (important)  technical details are missing.

1.       The description of state-of-the-art methods is very scarce. References are cited, but no details are given as to their content/results.

2.       The description of the model in Figure 2 in lines 145-150 does not agree with the figure scheme. Also, be more precise, e.g., “several blocks of convolutional layers” – does it always means four blocks as indicated in the figure?

3.       Figure 3 is presented rather simplistically − first, we have some sequence, then a flat model that again returns a sequence, but we do not know how it maps to the input sequence. Next, we have more flat models returning a sequence of decisions − it would be helpful to distinguish which elements in the subsequent parts of the model follow directly from each other and, if they are independent, to indicate this as well. There is also a lack of information on how the final decision to assign to a particular class is made − in the structure presented, the model returns a sequence, which must then be analyzed somehow to make a binary decision.

4.       It seems that a more detailed description of the model training and the trained network will be valuable. This mainly concerns a more detailed description of the model’s architecture with the parameters of the layers, the number of parameters, the hardware on which these models were trained, the training time, the size of the “batch,” etc.

5.       Why were not newer collections such as Million Song Dataset, FMA, or other music genre contest-related datasets used? Also, there are larger datasets available than GTZAN, so for the deep model training, this should be taken into account (or commented on, at least).

6.       I could not find information on what division was used between training and the test collection. Is it “The filtered training set (443 tracks) is regarded as a part of the fusion training set”?

7.       Can you explain what it means: “It can achieve superior performance on the filtered GTZAN dataset” − I am referring to ‘filtered.’ If ‘filtered’ means including “difficult” examples, it would be useful to show what profit this gave on the same evaluation set.

8.       The analysis of the quality of the network, depending on the cost function, the type of input data, and the number of MLP layers, are valuable − this shows that the result is derived from a carefully conducted analysis. However, as said before, a detailed description of the model is a bit scarce.

9.       The results should refer to state-of-the-art. Obtaining “74.60% vs. 74.55%, 82.43% vs. 82.36” on the GTZAN dataset is not an outstanding achievement.

10.   Again, “In Figure 4, we observe that classic music and metal music are very well classified, while music with the genre of country and rock is harder to distinguish. Overall, the performance of the automatic classification system based on LGNet is satisfactory, it can make correct predictions for most of the music tracks. -> Such a statement should be compared to state-of-the-art.

Detailed remarks

- Be more careful about spaces and punctuation in the text, e.g., learning[1]. genres[2,3]. (MFCCs)[8,9,19]...

- Should references not be recalled as they appear in the text?

- rhymes –> rhythms?

- Not all acronyms are listed in the text, e.g., RNN-LSTM

- rhythm, and beats -> rhythm and beats

- The framework  of network backbone -> use another word (a technical term) instead of ‘backbone’

- with  classic classification -> a repetition, better use: baseline

- The overall Training flow -> The overall training flow

Author Response

Thanks for your constructive comments. Please see the attachment for our response.
